# Controllable Doping Characteristics for WS_x_Se_y_ Monolayers Based on the Tunable S/Se Ratio

**DOI:** 10.3390/nano13142107

**Published:** 2023-07-19

**Authors:** Chen Ji, Yung-Huang Chang, Chien-Sheng Huang, Bohr-Ran Huang, Yuan-Tsung Chen

**Affiliations:** 1Graduate Institute of Electro-Optical Engineering, Department of Electronic and Computer Engineering, National Taiwan University of Science and Technology, Taipei 106335, Taiwan; 2Bachelor Program in Industrial Technology, National Yunlin University of Science and Technology, Douliu 64002, Yunlin, Taiwan; 3Department of Electronic Engineering, National Yunlin University of Science and Technology, Douliu 64002, Yunlin, Taiwan; 4Graduate School of Materials Science, National Yunlin University of Science and Technology, Douliu 64002, Yunlin, Taiwan

**Keywords:** transition metal dichalcogenides, WS_x_Se_y_ monolayers, energy band diagrams, doping characteristics

## Abstract

Transition metal dichalcogenides (TMDs) have attracted much attention because of their unique characteristics and potential applications in electronic devices. Recent reports have successfully demonstrated the growth of 2-dimensional MoS_x_Se_y_, Mo_x_W_y_S_2_, Mo_x_W_y_Se_2_, and WS_x_Se_y_ monolayers that exhibit tunable band gap energies. However, few works have examined the doping behavior of those 2D monolayers. This study synthesizes WS_x_Se_y_ monolayers using the CVD process, in which different heating temperatures are applied to sulfur powders to control the ratio of S to Se in WS_x_Se_y_. Increasing the Se component in WS_x_Se_y_ monolayers produced an apparent electronic state transformation from p-type to n-type, recorded through energy band diagrams. Simultaneously, p-type characteristics gradually became clear as the S component was enhanced in WS_x_Se_y_ monolayers. In addition, Raman spectra showed a red shift of the WS_2_-related peaks, indicating n-doping behavior in the WS_x_Se_y_ monolayers. In contrast, with the increase of the sulfur component, the blue shift of the WSe_2_-related peaks in the Raman spectra involved the p-doping behavior of WS_x_Se_y_ monolayers. In addition, the optical band gap of the as-grown WS_x_Se_y_ monolayers from 1.97 eV to 1.61 eV is precisely tunable via the different chalcogenide heating temperatures. The results regarding the doping characteristics of WS_x_Se_y_ monolayers provide more options in electronic and optical design.

## 1. Introduction

Transition metal dichalcogenides (TMDs) have recently attracted considerable research attention due to their atomic monolayer structure, moderate carrier mobility [1,2], direct band gap [3,4,5], outstanding flexibility [6], and excellent optical properties [7,8]. These unique characteristics allow them to serve as flexible field effect transistors (FETs) [9,10,11,12], photovoltaic cells [13,14], light-emitting diodes [15,16], photodetectors [17], and catalysts [18,19,20]. In particular, TMD monolayers have shown up to 5–10% sunlight absorption ability [13], more than 2200 A/W photoresponsivity [21], and pronounced threshold behavior in electroluminescence [17], making them well suited for applications in optoelectronic devices. Optoelectronic performance, in areas such as efficiency and optical responsivity, is extremely dependent on the optical band gap of the semiconductor TMD materials. Therefore, a technology is urgently needed to control the optical band gap of TMD monolayers. Early strain engineering [22,23,24,25] and stacking of various TMD monolayers [26,27] have successfully modified the optical band gap to a limited degree. Recently, a 2H phase MoS_2_ monolayer was modified by 1T phase MoS_2_ quantum dot arrays through electron beam irradiation, displaying tunable band gap characteristics [28]. The synthesis technology for 2D TMD alloys is also an important engineering area for studying their band gap and tuning doping characteristics and structures. The selenization/sulfurization of as-grown MoS_2_/MoSe_2_ monolayers using selenium/sulfur to replace the original chalcogenide elements was proposed to form MoS_x_Se_y_ alloys [29]. Note that the selenization/sulfurization process makes it difficult to modulate the band gap at the assigned emission position. Although Mo_x_W_y_S_2_ and Mo_x_W_y_Se_2_ alloys have been acquired via mechanical exfoliation from bulk crystals [30,31,32], the flake structures would inhibit their development in future applications. Recently, a series of 2D TMD alloys (e.g., MoS_x_Se_y_, Mo_x_W_y_S_2_, Mo_x_W_y_Se_2_, and WS_x_Se_y_) has been prepared by CVD, with tunable band gap energies controlled via chemical compositions [33,34,35,36,37,38,39,40,41]. However, most of the studies in TMD alloys were focused on fabrication, optical analysis, TEM investigation, electricity, band gap energies, and theoretical exploration. Few studies in the literature have discussed the transition of electronic properties via the changes in the concentration of transition metals/chalcogenides. Even energy band diagrams as a function of transition metal/chalcogenide concentration in TMD alloys are not explored via ultraviolet photoemission spectroscopy (UPS). Duan et al. used this approach to directly prepare WS_2x_Se_2−2x_ monolayers by using different ratios of WS_2_ and WSe_2_ powders mixed together in the CVD process [34]. In addition, increasing the S element in WS_2x_Se_2−2x_ monolayers resulted in the transition of electronic properties from p-type WSe_2_ to n-type WS_2_ by back-gated field effect transistors. However, the calculations of density functional theory claim that WS_2_ was p-type and would be transformed into n-type WS_x_Se_y_ when enough Se element was added to the WS_x_Se_y_ [42]. These inconsistent conclusions point to the urgent need to systematically explore the doping behaviors of these 2D monolayers.

In this study, we report the synthesis of WS_x_Se_y_ monolayers using tungsten oxides, selenium, and sulfur powders as the sources for the CVD process, in which different heating temperatures are applied to sulfur powders for S/Se ratio modulation. The optical band gap of the as-grown WS_x_Se_y_ monolayers from 1.97 eV (WS_2_) to 1.61 eV (WSe_2_) was precisely controlled through the tunable S/Se ratio. The red shift of the WS_2_-related peaks in Raman spectra involved n-doping behavior in WS_x_Se_y_ monolayers, whereas the blue shift of the WSe_2_-related peaks indicated p-doping characteristics. The electronic state transformation of WS_x_Se_y_ monolayers could be tuned from p-type WS_2_ toward n-type WSe_2_ by systematically controlling the S/Se ratio as recorded through UPS measurements. The reported observations of the doping characteristics in these WS_x_Se_y_ monolayers have implications for electronic and optical design [17,43,44,45].

## 2. Experimental Section

### 2.1. Synthesis of Monolayer WS_2_ and WSe_2_

Crystal WS_2_ and WSe_2_ triangles were synthesized by modifying the processes described in our previous work. In brief, the WO_3_ powders (0.3 g; Sigma-Aldrich, New Taipei City, Taiwan, 99.5%) were placed in a quartz boat located in the heating zone center of the furnace. The S powders (Sigma-Aldrich, 99.5%) were placed in a separate quartz boat on the upper stream side. The sapphire substrates for growing WS_2_ were located downstream close to the WO_3_ powders. The central heating zone was first heated to 500 °C at 10 °C/min with an Ar/H_2_ flowing gas (Ar = 70 sccm, H_2_ = 5 sccm, chamber pressure = 5 Torr) and kept for 20 min. The furnace was then heated to 925 °C at a ramping rate of 25 °C/min and kept for 15 min. The sulfur was heated separately by a heating belt to 160 °C when the furnace reached 650 °C. After growth, the furnace was slowly cooled to room temperature. For WSe_2_ growth, the Se powders (Sigma-Aldrich, 99.5%) were substituted for S powders as the source of the Se element, and the same growth process was followed, except for the heating belt being heated to 270 °C during growth.

### 2.2. Synthesis of Monolayer WS_x_Se_y_ Alloys

To synthesize WS_x_Se_y_, the sulfur and selenium powders were separately but simultaneously placed in two quartz boats on the upper stream side. During WS_x_Se_y_ alloy growth, the selenium powders were continuously heated to 260 °C by a heating belt and, for the various proportion of ingredients, the heating temperature of the sulfur powders was increased from 80 °C, 90 °C, 100 °C, and 110 °C to 120 °C. The location of the WO_3_ powders (0.3 g) and the growth procedure are identical to the conditions previously described. After growth, the furnace was allowed to cool to room temperature.

### 2.3. Characterizations

Raman spectra were collected with an NT-MDT confocal Raman microscopic system (laser wavelength 473 nm and laser spot size ~0.5 μm). The Si peak at 520 cm^−1^ was used as a reference for wavenumber calibration. The atomic force microscope (AFM) images were performed using a Veeco Dimension Icon system. The photoluminescence (PL) and differential reflectance spectra were measured with a homemade microscopy system. For PL measurements, a 532 nm solid-state laser was focused to a spot size < 1 μm on the sample by an objective lens (×100; N.A. = 0.9). The PL signals were collected through the same objective lens, analyzed by a 0.75 m monochromator, and detected by a liquid-nitrogen-cooled charge-coupled device (CCD) camera. The apparatus for differential reflectance measurements was basically the same, except that the light source was replaced by a fiber-coupled tungsten–halogen lamp. Chemical configurations were determined by X-ray photoelectron spectroscope (XPS, Phi V5000, Kanagawa, Japan). XPS measurements of the samples were performed with an Mg Kα X-ray source. The energy calibrations were made against the C 1 s peak to eliminate sample charging during analysis.

## 3. Results and Discussion

### 3.1. WS_x_Se_y_ Monolayer Fabrication

The growth of crystalline WS_2_ and WSe_2_ monolayers has been reported in our previous studies. Briefly, the triangular WS_2_ and WSe_2_ flakes are fabricated through the vapor phase reaction of WO_3_ with S and Se powders, respectively; similar methods have been demonstrated by many other groups [46,47,48]. The experimental set-up for as-deposited WS_x_Se_y_ monolayers in a hot-wall furnace is illustrated in Figure 1a. To synthesize the WS_x_Se_y_ monolayers, S and Se powders were introduced simultaneously into the furnace during the growth process. Moreover, the Se powder was maintained at 260 °C, while the S powder was heated incrementally from 80, 90, 100, and 110 to 120 °C to modulate the S/Se ratio, labeled as Ts = 80 °C, Ts = 90 °C, Ts = 100 °C, Ts = 110 °C, and Ts = 120 °C. Figure 1c–e show the optical micrographs (OM) of the WS_x_Se_y_ monolayers on sapphire substrates, which are at different locations far from the WO_3_ precursor. First, Figure 1c shows sparsely isolated triangular flakes at the farthest place from the WO_3_ precursor, indicating that the nucleation density and the precursor density are low. Close to the WO_3_ precursor, small isolated triangles grew generally and then merged together with a lateral size of ~50 µm, shown in Figure 1d. Finally, due to the enlarged number of precursors, the WS_x_Se_y_ domains closest to the WO_3_ precursor merged together and formed a continuous complete film. Some bilayer flakes are still observed on the top of the continuous monolayer film, which is attributed to the nucleation assisted by the grain boundary or particles, as shown in Figure 1e. In addition, according to the cross-sectional height of ~0.85 nm, inspected by atomic force microscope (AFM) in Figure 1b, the monolayer structure of the WS_x_Se_y_ flake was confirmed [30].

Figure 2a shows the normalized photoluminescence (PL) spectra for the WS_2_, WSe_2_, and WS_x_Se_y_ monolayers. The PL peak positions for the pristine WS_2_ and WSe_2_ are, respectively, located at 2.0 and 1.64 eV [49], attributed to direct emission from the conduction band minimum (CBM) to the valence band maximum (VBM) for A excitons at the K point in the Brillouin zone [3,50]. When the heating temperature of the S powder was reduced from Ts = 120, 110, 100, and 90, to 80 °C, the PL peak positions of the WS_x_Se_y_ monolayers gradually decreased from 1.91, 1.88, 1.83, and 1.76 to 1.70 eV, respectively. Therefore, a tuneable PL emission position can be easily achieved based on the modulation of the S/Se ratio in WS_x_Se_y_ monolayers by controlling the heating temperatures of the S and Se precursors. The strong emission from A excitons without B excitons for WS_x_Se_y_ compounds is in good agreement with the direct band gap emission in a monolayer, consistent with pristine TMD monolayer materials [51]. Furthermore, the only strong PL peak observed in the WS_x_Se_y_ monolayers indicates a uniform distribution of S and Se in the compound domain. Otherwise, two apparent characteristic peaks, respectively belonging to MoS_2_ and MoSe_2,_ would present simultaneously in the PL spectrum due to distinguishable components of MoS_2_ and MoSe_2_ in the WS_x_Se_y_ monolayer, shown in Appendix A. Figure 2b shows the optical absorption spectra for these WS_2_, WSe_2_, and WS_x_Se_y_ monolayers. Two distinct A and B excitonic absorption peaks for WS_2_ (WSe_2_) monolayers are at 2.01 and 2.40 eV (1.65 and 2.07 eV), respectively, resulting from the spin–orbital splitting of the valence band [52]. The two A and B absorption peaks for the WS_x_Se_y_ monolayers at Ts = 120, 110, 100, and 90, to 80 °C are located at 1.93 and 2.33 eV, 1.89 and 2.29 eV, 1.84 and 2.26 eV, and 1.79 and 2.22 eV to 1.73 and 2.14 eV, respectively. An absorption spectrum represents the energy required for electrons to be excited from the valence band (E_V_) to the conduction band (E_C_). Therefore, the optical band gap energy (E_g_) can be determined from the absorption coefficient near the absorption edge shown in Appendix A (as described in the previous study) [53]. The optical band gap energy of the WS_2_, WS_x_Se_y_ at Ts = 120, 110, 100, 90, and 80 °C, and WSe_2_ monolayers is 1.97, 1.88, 1.85, 1.79, 1.75, 1.67, and 1.61 eV, respectively. Hence, the optical band gap energy of WS_x_Se_y_ alloys could be controlled precisely in the range between that of the WS_2_ and WSe_2_ monolayers.

### 3.2. Raman Characterizations

The Raman spectra of as-deposited WS_2_, WSe_2_, and WS_x_Se_y_ monolayers on sapphire substrates are shown in Figure 3a. Pristine WS_2_ shows two distinct characteristic peaks of $E_{2g}^{1}$ and A_1g_ at 359.8 and 420.7 cm^−1^, respectively, due to the in-plane and out-of-plane vibration models of the atoms [49]. The characteristic peaks of $E_{2g}^{1}$ and A_1g_ at 251.7 and 262.7 cm^−1^ for pristine WSe_2_ were also confirmed [49]. However, in the case of the WS_x_Se_y_ monolayers, both WS_2_-related and WSe_2_-related characteristic peaks were simultaneously observed in the spectrum. In addition, the shifts of the WS_2_-related and WSe_2_-related characteristic peaks were seen in opposite directions. Compared with pristine WS_2_, a slight red shift for the WS_2_-related peaks of $E_{2g}^{1}$ and A_1g_ at 358 and 415.7 cm^−1^ was revealed after Se doping in pristine WS_2_ at the Ts = 120 °C stage. Meanwhile, the intensity of the WS_2_-related $E_{2g}^{1}$ peak decreased dramatically, and full width at half maximum (FWHM) was also broadened in both WS_2_-related $E_{2g}^{1}$ and A_1g_ peaks. Note that an unidentified peak at 265 cm^−1^ could be attributed to the vibration from the degenerate $E_{2g}^{1}$ mode of W-Se structures. Furthermore, as the amount of selenium in the WS_x_Se_y_ monolayers is increased by the Ts = 120 °C to Ts = 80 °C processes, the WS_2_-related A_1g_ peak shows a red shift trend from 415.7, 413.8, 412.7, and 407 to 406 cm^−1^. However, the position of the WS_2_-related $E_{2g}^{1}$ mode does not clearly change except for the enlarged FWHM associated with the relaxation of the Raman selection rule at defects [54,55]. Although stretching strain caused a red shift [56], compressive strain produces the opposite blue shift [22]. Due to the larger lattice constant of a = 3.25 Å for WSe_2_ (a = 3.13 Å for WS_2_) [57], the WS_2_ monolayer would suffer compressive strain arising from Se atom doping. Therefore, the red shift of the WS_2_-related peaks does not result from the compressive strain arising from Se atom doping in the WS_x_Se_y_ monolayers. The increased red shift of the WS_2_-related peaks may be attributed to the effect of changing carrier concentrations on phonon vibrations arising from the increased amount of selenium in the WS_x_Se_y_ monolayers, where relevant investigations have been reported on the Au nanoparticle-decorated MoS_2_ [48,53] and MoS_2_/graphene stacks [58]. In addition to WS_2_-related peaks, a blue shift of WSe_2_-related $E_{2g}^{1}$ at 259.7 cm^−1^ was also observed for sulfur doping in pristine WSe_2_ at the Ts = 80 °C stage. Furthermore, as the amount of sulfur in the WS_x_Se_y_ monolayers increases from Ts = 80 °C to Ts = 120 °C, the WSe_2_-related $E_{2g}^{1}$ peak shows an expanded blue shift, broadened FWHM, and decreased intensity, as shown in Figure 3a. Interestingly, the opposite shift direction for the WS_2_- and WSe_2_-related Raman peaks was apparently revealed, suggesting that doping behaviors for S and Se atoms in WS_x_Se_y_ monolayers may result in different changes in carrier concentrations or strain. However, as previously mentioned, according to the smaller lattice constant for WS_2_ [57], the blue shift of the WSe_2_-related peaks does not result from the stretching strain arising from S atom doping in the WS_x_Se_y_ monolayers. Hence, the strain is not the main reason for the shifts of both WS_2_- and WSe_2_-related peaks in the Raman spectra, indicating that the carrier concentration is the key factor. Consequently, according to prior work [22,56,59], the red shift of WS_2_-related Raman peaks through the increase of the Se element in WS_x_Se_y_ monolayers may be associated with the change of carrier concentration toward the n-type, while the blue shift of WSe_2_-related Raman peaks via the S atom increase in WS_x_Se_y_ monolayers could involve an increase in hole carrier concentrations. The doping behaviors for carrier concentration will be discussed further below. A multilayer-related peak at 307 cm^−1^ is not observed, indicating that these WS_x_Se_y_ materials are monolayers [47]. The Raman shifts referenced to pristine characteristic peaks and FWHM for the WS_2_-related $E_{2g}^{1}$ and A_1g_ peaks and the WSe_2_-related $E_{2g}^{1}$ peak at various heating temperatures are shown in Figure 3b,c.

### 3.3. PL and Raman Mapping

To confirm the homogeneity of the WS_x_Se_y_ monolayers, an optical micrograph and the corresponding PL and Raman mapping of a triangular WS_x_Se_y_ flake are shown in Figure 4. Figure 4a shows the isolated monolayer WS_x_Se_y_ triangle with a lateral size of ~10 µm, grown using the Ts = 120 °C process. The corresponding peak intensity and position mappings of PL, WS_2_-related A_1g_ Raman mode, and WSe_2_-related $E_{2g}^{1}$ Raman mode show homogeneous intensity within the same individual domains, respectively shown in Figure 4b–d and Figure 4e–g. However, the slight variations of PL position mapping in Figure 4c changing from 650 nm to 635 nm (~45 meV) could be attributed to componential fluctuations within the triangular flake, consistent with prior findings [60]. No shift in PL position at the edges of the triangles was found, suggesting no strain effects on the edges from the substrates [60,61]. However, the remarkable suppression of PL intensity at the edges shown in Figure 4b results from edge-localized states in the band gap, structural imperfections, or charged defects that quenched the PL [60]. Note that the PL intensity for the WS_x_Se_y_ monolayer is much stronger than the Raman signal, indicating superior crystallinity and lower defects in the as-grown WS_x_Se_y_ monolayers. In addition, the homogeneous Raman intensity mappings shown in Figure 4d,f present excellent uniformity of crystalline quality within the WS_x_Se_y_ flake. The tiny variation within ±2 cm^–1^ for both Raman position mappings in Figure 4e,g show that WS_2_-related and WSe_2_-related materials were uniformly distributed over the whole WS_x_Se_y_ flake, indicating good mixing and consistent components for the W, S, and Se elements. Therefore, the mapping results show that WS_2_-related and WSe_2_-related materials are uniformly mixed to form an homogeneous alloy monolayer.

### 3.4. X-ray Photoemission Spectroscopy (XPS)

The surface composition and stoichiometry of the as-deposited WS_2_, WSe_2_, and WS_x_Se_y_ monolayers were characterized using X-ray photoemission spectroscopy (XPS). Figure 5 shows pristine WS_2_ with two characteristic peaks at 32.8 and 35.0 eV, attributed to the doublet W 4f_7/2_ and W 4f_5/2_ binding energies for W^4+^, whereas WSe_2_ presents a slight red shift at 32.1 and 34.3 eV due to weaker electronegativity [47]. The doublet peaks corresponding to the S 2p_3/2_ and S 2p_1/2_ orbital of divalent sulfide ions (S^2−^) are observed at 162.1 and 163.3 eV [29]. The doublet peaks for Se^2−^ at 54.3 and 55.1 eV are assigned to the Se 3d_5/2_ and Se 3d_3/2_ binding energies [29,47]. In addition, the weak doublet peaks associated with WO_3_ at 35.8 and 38.0 eV are also observed in all samples, possibly resulting from incomplete-reaction WO_3_ precursors or oxidation from residual oxygen in the chamber [47]. When the sulfur heating temperature decreases, the sulfur doublet peaks (S 2p_3/2_ and S 2p_1/2_) gradually become less evident, while the two selenium doublet peaks (Se 3d_3/2_, Se 3d_5/2_) and (Se 3p_3/2_, Se 3p_5/2_) become more prominent. The magnitude of each profile was normalized for easier comparison. By changing the sulfur heating temperature from 120 to 80 °C, various stoichiometries from the WS_1.87_Se_0.31_ to WS_0.88_Se_1.39_ for WS_x_Se_y_ monolayers could be controlled precisely, specifically listed in Appendix A. Hence, various concentrations of S and Se elements in the WS_x_Se_y_ monolayers could be accurately modulated by controlling the precursor heating temperature. In addition, the chemical stoichiometry of these WS_x_Se_y_ monolayers is chalcogen-plentiful; that is, the ratio of (S + Se)/Mo is greater than 2, which could be attributed to lower WS_x_Se_y_ formation enthalpies evaluated by first-principle calculations [62] or excess chalcogen elements in the process.

### 3.5. Ultraviolet Photoemission Spectroscopy (UPS)

The energy level alignment with respect to the Fermi energy (E_F_) was explored using ultraviolet photoemission spectroscopy (UPS). The Au layer was used as a reference to ensure that the Fermi energy was located at 0 eV [53]. The valence band below the E_F_ (E_F_-E_V_) for WS_2_, WS_x_Se_y_ at Ts = 120, 110, 100, 90, and 80 °C, and WSe_2_ monolayers is 0.735, 0.835, 0.885, 0.90, 0.91, 0.91, and 0.96 eV, respectively, acquired by linearly extrapolating the leading edge of the spectrum to the baseline shown in Figure 6a. Moreover, the work function (Φ) can be estimated using Φ = hν − E_onset_, where hν is the incident photon energy (21.2 eV) and E_onset_ is the onset level related to the secondary electrons, as shown in Figure 6b [53]. Thus, the Φ for WS_2_, WS_x_Se_y_ at Ts = 120, 110, 100, 90, and 80 °C, and WSe_2_ monolayers is 4.31, 4.17, 4.06, 4.03, 4.00, 3.96, and 3.95 eV, respectively. In addition, the optical band gaps of the WS_2_, WS_x_Se_y_, and WSe_2_ monolayers were discussed above in Appendix A. The values of the optical band gap, Φ, and E_F_-E_V_ for the WS_2_, WS_x_Se_y_, and WSe_2_ monolayers are listed in Appendix A. The energy band diagrams relative to the E_F_ for the WS_2_, WS_x_Se_y_, and WSe_2_ monolayers are shown in Figure 6c. The E_F_-E_V_ energy of the WS_2_ is 0.735 eV, which is smaller than half the band gap energy, indicating that the WS_2_ monolayer is a p-type semiconductor material, consistent with other reports [63,64]. With an increase of the Se component in the WS_x_Se_y_ monolayers (i.e., a decrease of Ts), the E_F_-E_V_ energy increased, and the E_F_ moved from the valence band toward the conduction band, demonstrating the transformation of electronic states from p-type to n-type. In addition, n-type WSe_2_ was identified according to the E_F_ position near to the conduction band, consistent with other reports [65,66]. The band gap and the energy band diagrams as a function of Se concentration in WS_x_Se_y_ monolayers are, respectively, in Figure 6d,e. Good linear fitting for the band gap and Se concentration in Figure 6d suggested that the band gap could be determined precisely and linearly through the control of Se concentration in WS_x_Se_y_ monolayers. In addition, the electronic state model could be tuned to p-type or n-type by modulating the Se concentration in the WS_x_Se_y_ monolayers. Therefore, the WS_x_Se_y_ monolayers can be adjusted as p-type or n-type semiconductors by systematically modulating the S/Se ratio in the process.

## 4. Conclusions

WS_x_Se_y_ monolayers were synthesized using tungsten oxides, selenium, and sulfur powders as the sources in the CVD process, in which different heating temperatures for the selenium and sulfur powders are applied, respectively, to control the S/Se ratio. The tunable band gap of the as-grown WS_x_Se_y_ monolayers changed from 1.97 eV to 1.61 eV with different chalcogenide heating temperatures, consistent with findings in other literature of 626.6 nm to 751.9 nm. The red shift for WS_2_-related Raman peaks arising from an increase of the Se element in the WS_x_Se_y_ monolayers was associated with an increase in electron concentration, whereas the blue shift for the WSe_2_-related Raman peaks was related to enhanced hole concentration. The homogeneous element distribution within a WS_x_Se_y_ flake was identified by PL and Raman mapping. The chemical stoichiometry for the WS_2_, WS_x_Se_y_ at Ts = 120, 110, 100, 90, and 80 °C, and WSe_2_ monolayers was, respectively, WS_2.20_, WS_1.87_Se_0.31_, WS_1.66_Se_0.40_, WS_1.54_Se_0.48_, WS_1.12_Se_1.00_, WS_0.88_Se_1.39_, and WSe_1.77_, indicating good control of the S/Se ratio via the chalcogenide heating temperature. With an increase of the Se element in the WS_x_Se_y_ monolayers, the work function changed from 4.31, 4.17, 4.06, 4, and 3.96 to 3.95, demonstrating the electronic state transition from p-type to n-type. The study of doping characteristics in those WS_x_Se_y_ monolayers via different chalcogen heating temperatures provides useful implications for electronic and optical design.

## Figures and Tables

**Figure 1 nanomaterials-13-02107-f001:**
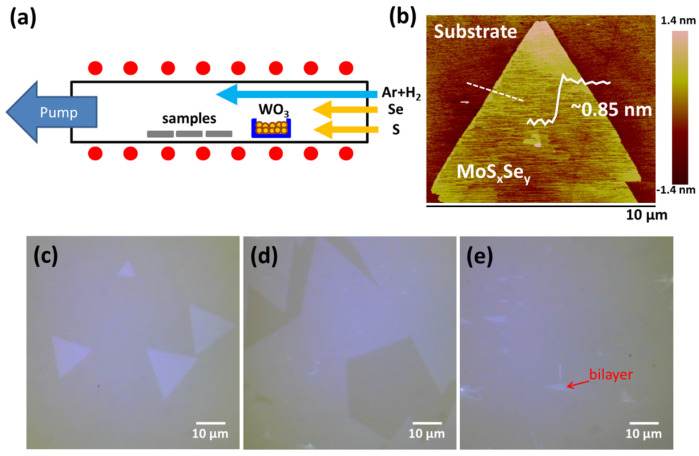
(**a**) Schematic illustration for the growth of WS_x_Se_y_ monolayers on sapphire substrates in a CVD furnace. (**b**) The AFM image of the WS_x_Se_y_ monolayers. (**c**–**e**) Optical micrograph of the monolayer WS_x_Se_y_ flakes and film, where the difference is the location of the substrates in the furnace.

**Figure 2 nanomaterials-13-02107-f002:**
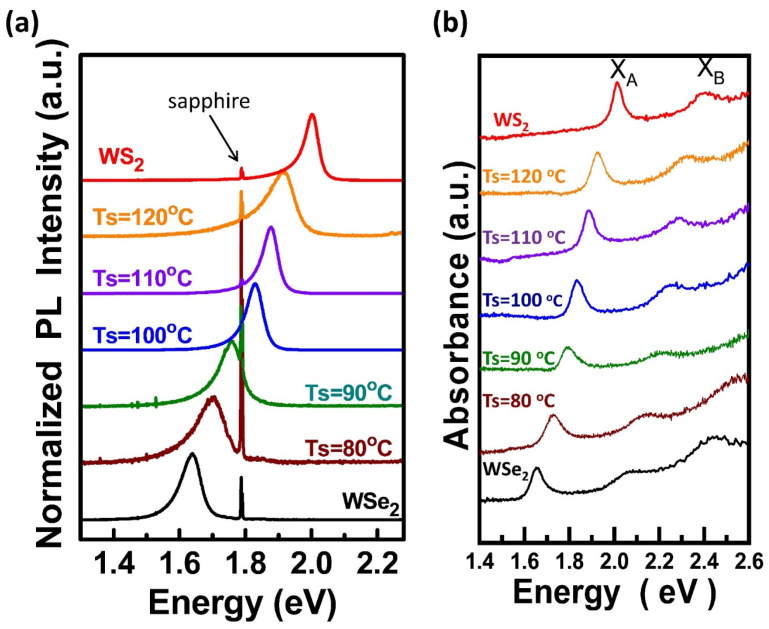
(**a**) Photoluminescence spectra for the WS_2_, WS_x_Se_y_ at Ts = 120, 110, 100, 90, and 80 °C, and WSe_2_ monolayers. (**b**) Optical absorption spectra for the WS_2_, WS_x_Se_y_ at Ts = 120, 110, 100, 90, and 80 °C, and WSe_2_ monolayers.

**Figure 3 nanomaterials-13-02107-f003:**
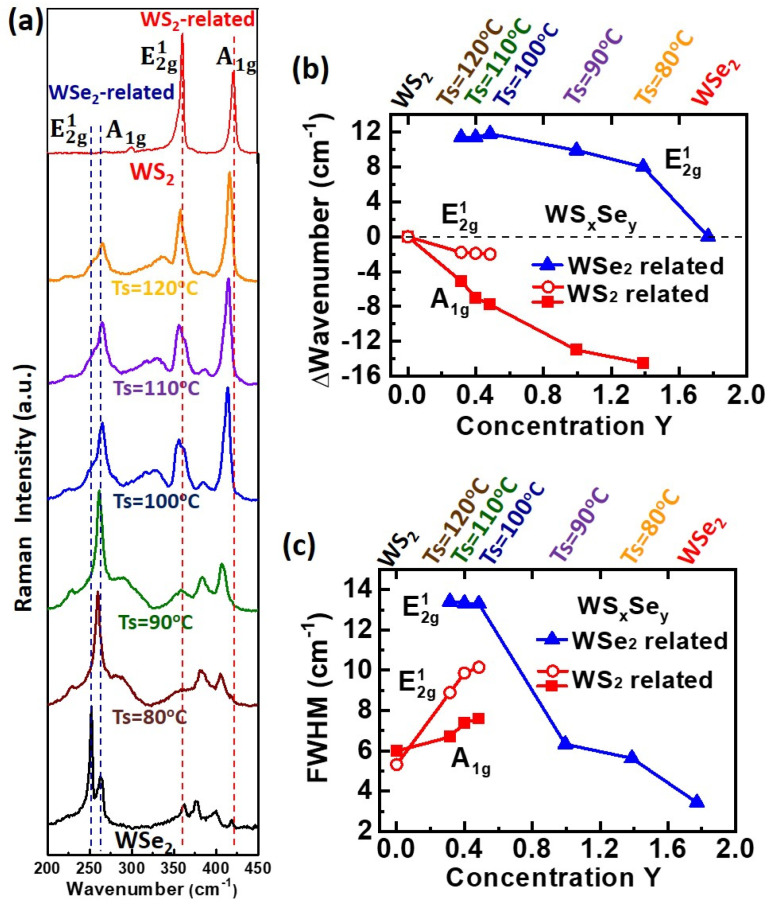
(**a**) Raman spectra for the WS_2_, WS_x_Se_y_ at Ts = 120, 110, 100, 90, and 80 °C, and WSe_2_ monolayers. The (**b**) Δwavenumber and (**c**) FWHM of WS_2_-related A_1g_ and $E_{2g}^{1}$ mode and WSe_2_-related $E_{2g}^{1}$ mode as a function of Se stoichiometry concentration in the WS_x_Se_y_ monolayers. The Δwavenumber is the Raman shift referenced to pristine characteristic peaks.

**Figure 4 nanomaterials-13-02107-f004:**
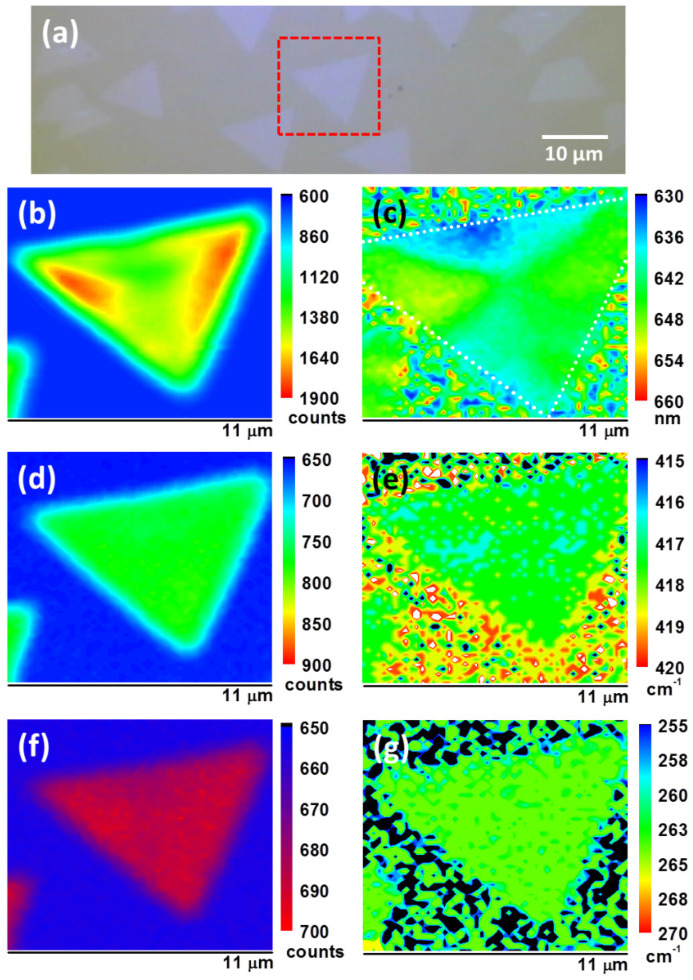
(**a**) Optical micrograph of the monolayer WS_x_Se_y_ flakes grown in the Ts = 120 °C process. Typical PL (**b**) intensity and (**c**) position mappings of the isolated WS_x_Se_y_ monolayer. Raman (**d**) intensity and (**e**) position mappings for the WS_2_-related A_1g_ mode at 415.7 cm^−1^. Raman (**f**) intensity and (**g**) position mappings for the WSe_2_-related $E_{2g}^{1}$ mode at 265 cm^−1^.

**Figure 5 nanomaterials-13-02107-f005:**
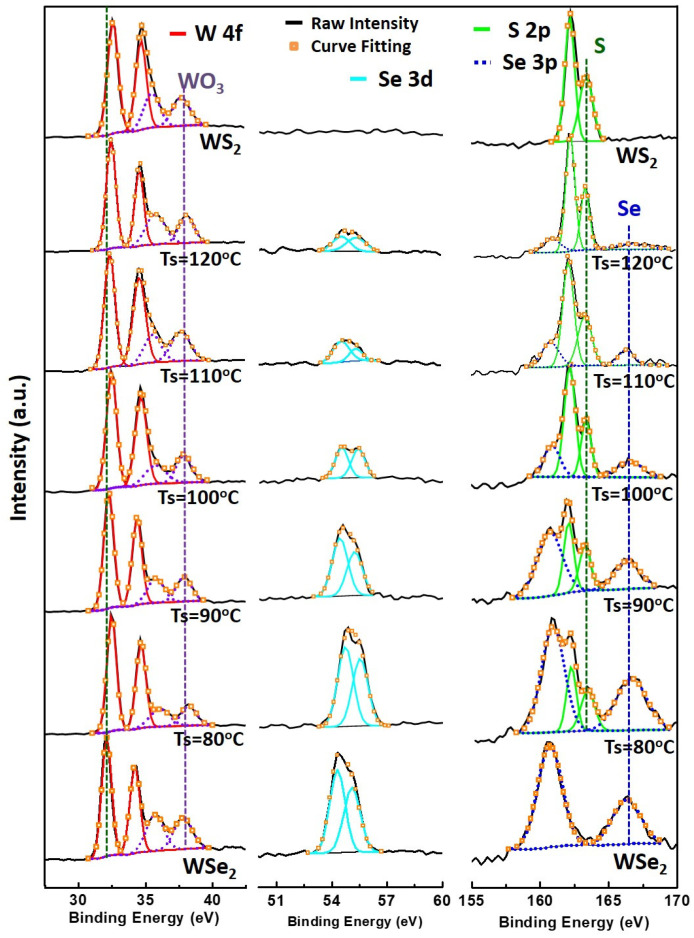
XPS scans for the W, S, and Se binding energies for the WS_2_, WS_x_Se_y_ at Ts = 120, 110, 100, 90 and 80 °C, and WSe_2_ monolayers. The magnitude of each profile is normalized for easier comparison.

**Figure 6 nanomaterials-13-02107-f006:**
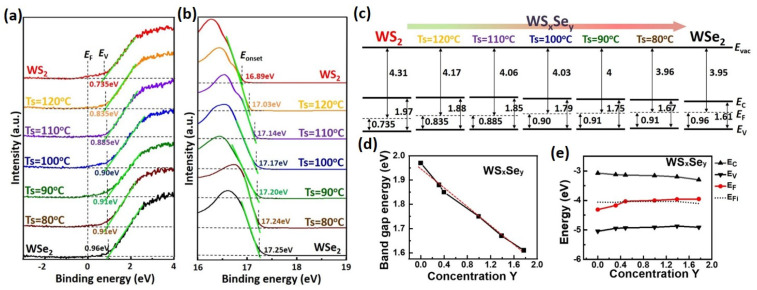
Ultraviolet photoemission spectroscopy and energy band diagrams. (**a**) UPS spectra near the Fermi level energy and valence band maximum of the monolayer WS_2_, WS_x_Se_y_ at Ts = 120, 110, 100, 90, and 80 °C, and WSe_2_ film transferred onto the 60 nm Au-coated Si substrates. (**b**) The onset level (E_onset_) of the UPS spectra, where the work function (Φ) can be calculated by Φ = hν − E_onset_. Here, hν is the incident photon energy of 21.2 eV. (**c**) The energy band diagrams of the WS_2_, WS_x_Se_y_ at Ts = 120, 110, 100, 90, and 80 °C, and WSe_2_ monolayers. (**d**) The band gap and (**e**) the energy band diagram as a function of Se stoichiometry concentration in the WS_x_Se_y_ monolayers.

## Data Availability

Research data are not shared.

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
