# Peer review of "Controllable Doping Characteristics for WSxSey Monolayers Based on the Tunable S/Se Ratio"

_nanomaterials, 2023, doi:10.3390/nano13142107_

Round 1
Reviewer 1 Report
This manuscript with the title “Controllable Doping Characteristics for WSxSey Monolayers Based on the Tunable S/Se Ratio” is submitted to journal Nanomaterials. The subject of this work is suitable for Nanomaterials. The WSxSey monolayers in this work are synthesized from the CVD process. To control the ratio of S and Se in the monolayers, heating temperatures of S and Se powders are used. When the Se concentration is increased, the change from p-type to n-type was obtained via energy band diagrams. And vice versa, with the S concentration increasing, the p-type was obtained.
The results for the whole range of monolayers from WSe2 to WS2, i.e. the corresponding temperature range of 80-120 degr C for the monolayers, are obtained using Raman spectroscopy, X-ray photoemission spectroscopy (XPS), and ultraviolet Photoemission Spectroscopy (UPS). The results are detailed and consistent. It is checked that the WS2 and WSe2 materials are uniformly mixed together and become a homogeneous alloy monolayer. The authors demonstrated that the applied techniques of heating provide a reliable way to control the optical band gap of the WSxSey monolayers from 1.97 to 1.61 eV with the concentration of S/Se in the WSxSey monolayers. Which is a great tunable opportunity for applications of these materials. The conclusions are fully supported by the main text and supplementary materials.
I have two issues about the manuscript:
1. References 43-45, 47 should be discussed in more details in the introduction section.
2. It is not clear from the text, how the x concentration changes with the temperature. Because most dependencies, Figs 4b,c and Figs 6d,e are plotted as “Concentration Y”.
The manuscript can be published in Nanomaterials.
Minor English editing is required
Reviewer 2 Report
The paper reports on the growth by CVD of WSSe alloy with variable composition of S and Se components. In particular, Authors use a specific furnace configuration, already presented elsewhere, and control composition by acting on the temperature.
Produced samples are analyzed by a range of conventional methods, including PL, Raman, XPS. Data show that the bandgap energy, affected by composition, can be modified. This is an interesting result, certainly worth to be published.
However, prior to acceptance of the manuscript Authors are requested to consider the following points and to prepare a revised version, accordingly.
1. Style must be definitely improved. In many instances inappropriate language is used and, in some cases, choice of terms does not appear to be compatible with a scientific publication. Authors are invited to revise the style with the help of a native English reviewer.
2. The topic of 2D materials has attracted strong interest in the last decade(s). As a consequence, related literature is very broad. For instance, most of the components of the paper, e.g., the fabrication method based on CVD, the occurrence of alloying and its role in determining optical properties, cannot be considered original. Originality (and motivations) of the work must be better pointed out in the Introduction. In particular Authors must underline in the text which is the gap of knowledge in the present state of the art their work aims to fill.
3. Within this frame, I note that also ties with previous, already published, works by the same team and the novel content of the present paper are difficult to be understood by the reader. This must be clarified in the text. Note also, that quotation of refs. 1-6 at line 118-119 appears not appropriate and must be corrected.
4. Moreover, a comparison between the present results and those acquired in similar systems (I mean, chalcogenides in general) in terms of bandgap control should appear in the Conclusions, in order for the reader to better catch and assess relevance of the paper.
Quality of the language is poor. While being correct from the point of view of syntax (and, although not always, grammar), sentence structure and, mostly, choice of terms sound not adequate for a scientific publication to appear in an international journal.
Reviewer 3 Report
The manuscript is well-structured and provide valuable information for readers. Please leave your comments for the following questions:
1- Please explain the reason for preheating of sulfur and Selenium to 160 C and 260 C by heating belt. does preheating process conducted in controlled atmosphere inside of CVD chamber?
2- The Si peak at 520 cm-1 was used 104 as a reference for wavenumber calibration in Raman , In the case of CVD process the sapphire was used as substrate. Does it mean you first calibrate your machine with the Bare Se substrate and later measured the films on sapphire substrates?
3- In line 106 please first write the Photoluminescence (PL) at first place and then use the abbreviated forms. Similar to other cases, CCD Camera.
4-Does the characterization process conducted under controlled atmosphere or in ambient condition?
